# Multi-View Modeling for Stock Investment Risk Forecasting

## Abstract

Forecasting stock investment risk is crucial for effective financial decision-making. Existing research on stock risk forecasting is still limited due to the lack of large-scale datasets and standardized investment risk forecasting tasks. To address this problem, we construct a stock investment risk dataset that standardizes the stock risk forecasting task as regression and classification problem, providing a benchmark for stock investment risk forecasting. Recent works only based on time series data capture a limited aspect of historical stock price data. To address this issue, we propose a multi-view framework that leverages large language models (LLMs) and pre-trained vision models to extract complementary long-periodic patterns and short-periodic patterns from historical stock data. Experimental results on our dataset demonstrate that our proposed model outperform the competitive baselines in regression task and classification task of stock investment risk forecasting. The codes and dataset are release in `https://anonymous.4open.science/r/MultiV-RF-F87F`.

## 1 Introduction

Portfolio allocation is necessary for investors to maximize their profits, where investors select a subset of stocks from the investment universe and dynamically allocate capital among the selected assets to achieve periodic returns. To help the portfolio allocation, recent works focus on forecasting stock prices returns or trends using historical market data (P.H. & Rishad, 2020; Zou et al., 2022; Wang et al., 2024a), demonstrating the feasibility of prediction using machine learning methods. However, these studies neglect the downside risk and volatility of the stocks that is considered as stock investment risk. Stock risk forecasting focuses on quantifying the potential losses and uncertainties associated with holding a stock, providing essential guidance for risk management, portfolio optimization, and stable investment decisions.

Recent works have built several datasets and benchmarks for stock price forecasting (Farimani et al., 2021; Sinha et al., 2022; Cyranka & Haponiuk, 2023). However, few works focus on stock investment risk forecasting dataset. To fill this gap, Luo & Liu (2025) quantify stock investment risks and construct a dataset of stock investment risk forecasting. However, the dataset lacks a standardized task formulation and is limited in scale. To this end, we scale up the benchmark and formulate regression and classification tasks of stock investment risk forecasting as well, which can contribute to the community of financial analysis.

Traditional stock forecasting methods based on time-series modeling primarily focus on the direct patch approach of stock time-series curves struggling to capture long temporal dependencies and periodic information (Szydłowski & Chudziak, 2024; Bui et al., 2025; Zhao et al., 2025). To overcome this limitation, we transform the stock curves into 2D matrix that can compress the long-periodic information as an additional view to the existing methods. Together with patch methods we propose a multi-view framework of extracting and modeling multi-view features of the stocks that can capture the short-periodic and long-periodic patterns of stock curves, which is shown in Figure 1.

Recently, large-scale foundation models demonstrate strong representation learning capabilities, including large language models (LLMs) such as GPT-2 and LLaMA (Radford et al., 2019; Touvron et al., 2023), pre-trained vision or language models such as MAE, BERT, and T5 (He et al., 2022; Devlin et al., 2019; Raffel et al., 2023), as well as multimodal foundation models such as CLIP, LLaVA, and ALIGN (Radford et al., 2021; Liu et al., 2023; Li et al., 2021). We backbone our multi-view modeling to a LLM feature extractor and pre-trained vision model to obtain short-periodic representation and long-periodic representation,

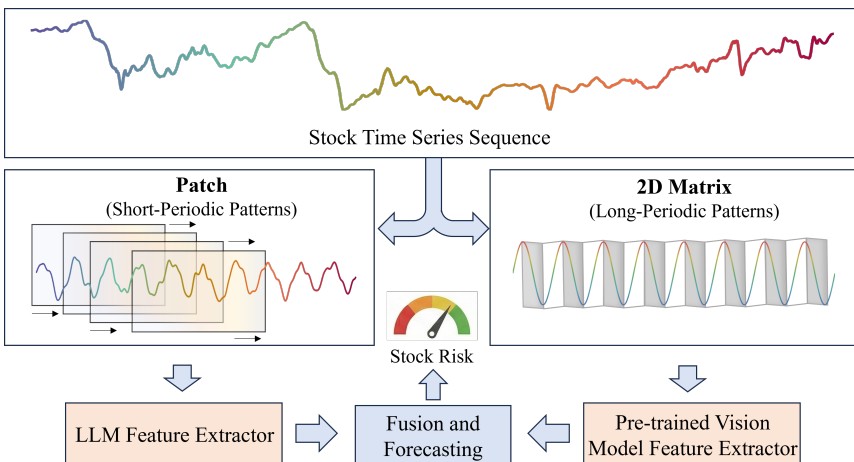

Figure 1: Multi-view modeling based on LLM feature extractor with short-periodic patterns and pre-trained vision feature extractor with long-periodic patterns.

respectively, which are mapped back to the temporal domain to form a unified representation for forecasting investment risk. The multi-view method significantly enhances the performance of stock investment risk forecasting. The contributions of our paper are summarized as follows:

- We build a large-scale comprehensive stock risk forecasting benchmark together with standard risk regression tasks and classification tasks, contributing to the scientific community of financial analysis.

- We propose a multi-view framework that leverages LLMs and pre-trained vision models to extract complementary short-periodic and long-periodic temporal features, providing rich representations for stock investment risk forecasting.

## 2 Related Work

### 2.1 Time Series Forecasting

**LLM-Based**  The development of LLMs inspires their application in time series forecasting. Their ability to capture dependencies and encode semantic structures provides a new paradigm for time series forecasting. Methods like Zhou et al. (2023) and Jin et al. (2023) transform time series into textual representation to forecast based on LLMs' reasoning ability. Pan et al. (2025) decomposes time series into time-domain and frequency-domain components to improve forecasting performance of LLMs. Pan et al. (2024) uses semantic informed prompt learning to improve LLM's ability of time series forecasting. Chang et al. (2023) proposes a two-stage alignment framework that adapts LLMs into data-efficient time-series forecasters by combining time-series alignment with forecasting fine-tuning. Liu et al. (2024b) propose token-wise prompting to preserving the LLM's contextual learning capabilities for time series forecasting. Talukder et al. (2024) proposes a tokenized representation of time series to learn general-purpose embeddings to address time series forecasting task. Ansari et al. (2024) introduces a framework that tokenizes time series into a vocabulary and pretrains transformer-based models on these tokens to create generalizable forecasting models.

**Image-Based**  Images have natural characteristic with time series that can provide structured representation for time series forecasting. Motivated by this observation, several recent works attempt to leverage image-based representations to improve forecasting accuracy. Wu et al. (2022) transforms time series into multi-periodic 2D data like images to capture time series dependencies. Wang et al. (2024b) transforms time series into multi-scale frequency-based time-image to enhance the model's ability to forecast temporal patterns. Chen et al. (2024) proposes to use pre-trained visual model to enhance time series forecasting performance.

## 2.2 Stock Time Series Forecasting

In traditional time series forecasting, the forecast targets are included in the historical observation space. But in the financial domain, stock time series forecast extends beyond the traditional tasks involving both targets outside the historical observation space, such as stock trends and stock rankings, as well as targets within it, such as stock prices. Several recent works have been proposed to handle stock time series forecasting with different target. Li et al. (2024) proposes a market-aware transformer framework to improve the accuracy and robustness of stock price forecasting. Liu et al. (2024a) proposes a time-aware multi-view learning framework about table data to improve the accuracy of stock rank forecasting. Zhu et al. (2024) presents a trend prediction model that captures both long- and short-term temporal dependencies using an enhanced GRU architecture.

## 2.3 Datasets for Stock Forecasting

Several datasets have been built for stock forecasting. Xu & Cohen (2018) combines historical stock prices and related tweets to facilitate prediction of stock price movements. FNSPID (Dong et al., 2024) is a large-scale dataset providing both the news content and temporal information for stock time series forecasting and analysis. QuanSIRA(Luo & Liu, 2025) is an investment risk dataset designed to forecast the risk level of stocks. However, comprehensive datasets and standardized risk forecasting task formulation for stock risk forecasting remain scarce, motivating our construction of a new dataset about stock investment risk forecasting.

# 3 Stock Investment Risk Benchmark

We introduce the benchmark with risk quantification and corresponding tasks of stock investment risk forecasting.

## 3.1 Data Sources

We collect stock data including adjusted close price, open price, high price, low price, and trading volume, for 200 companies in the S&P 500 from 1/1/2013 to 31/12/2023 using the yfinance API.[1] We train our model on this S&P dataset, as it is representative of a broad investment universe; detailed justification is provided in Appendix A.4.

## 3.2 Risk Indicator

We use three normalized risk indicators: volatility, maximum drawdown, and beta coefficient to quantify the investment risk of each stock according to Luo & Liu (2025).[2] The detailed normalization process in Appendix A.5.

**Volatility**  Price volatility is used to measure the changes in stock returns. High volatility represents stock uncertainty and low volatility represents stable stock. We quantify volatility, $V_t$ over the period $[t+1, t+\Delta d]$:

$$V_t = \sqrt{\frac{1}{\Delta d - 1} \sum_{i=t+1}^{t+\Delta d} (R_i - \bar{R})^2},$$

(1)

where $R_i$ is the return rate of a stock on $i$-th day:

$$R_i = \frac{p_i - p_{i-1}}{p_{i-1}},$$

(2)

---

[1] https://ranaroussi.github.io/yfinance/reference/index.html.
[2] The detailed distribution of the three normalized risk indicators is in Appendix A.1.

where $x_i$ is the closing price of a stock on $i$-th day, and $\bar{R}$ is the average return rate of a stock in the time $[t+1, t+\Delta d]$:

$$\bar{R} = \frac{1}{\Delta d} \sum_{i=t+1}^{t+\Delta d} R_i, \tag{3}$$

**Maximum drawdown**   Maximum drawdown is used to measure the greatest adverse fluctuation in the stock price over a specific period. Higher maximum drawdown represents a higher level of downside risk and lower maximum drawdown represents a lower level of downside risk. We quantify maximum drawdown, $M_t$ over the period $[t+1, t+\Delta d]$:

$$M_t = \max_{\tau \in [t+1, t+\Delta d]} \frac{P_{i,\max} - P_{i,\tau}}{P_{i,\max}}, \tag{4}$$

where $P_{i,\tau}$ denotes the price of stock $i$ at time $\tau$, $P_{i,\max}$ is the maximum price in period $[t+1, \tau]$.

**Beta coefficient**   Beta coefficient is used to measure the discrepancy between stock returns and market. Higher beta coefficient indicates that the asset reacts more strongly to market fluctuations and lower beta coefficient suggests weaker sensitivity to market movements, meaning lower systematic risk. To focus on the magnitude of sensitivity regardless of direction, we quantify the beta coefficient using its absolute value, denoted as $B_t$, over the period $[t+1, t+\Delta d]$:

$$B_t = \left| \frac{\text{Cov}(\bar{R}, \bar{R}_m)}{\sigma_m^2} \right|, \tag{5}$$

$$\sigma_m^2 = \frac{1}{\Delta d} \sum_{i=1}^{n} (R_i - \bar{R}_m)^2, \tag{6}$$

where $\bar{R}_m$ is the average yield rate of the market based on S&P 500.

### 3.3   Risk Quantification

We quantify the risk $S_t$ in $[t+1, t+\Delta d]$ as follow:

$$S_t = \frac{1}{3}(V_t + M_t + B_t), \tag{7}$$

where a larger $S_t$ indicates a higher risk for the corresponding stock.[3]   As shown in Figure 2, the distribution of the risk values $S_t$ shows a right-skewed and heavy-tailed behavior which is consistent with the risk distribution observed in real-world stock markets (Guo, 2017).

### 3.4   Tasks

Stock investment risk forecasting problem are categorized into regression tasks and classification tasks. The goal is to predict the future risk of a stock over the time window $[t+1, t+\Delta d]$, based on its historical observations from the

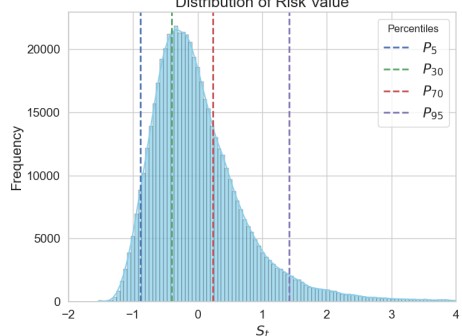

Figure 2: Distribution of the risk values $S_t$ across all samples in the dataset.

past time period $[t-w, t]$. The input is a time series feature $X \in \mathbb{R}^{w \times 1}$ from the stock historical data including adjusted close price, open price, high price, low price, or trading volume, where $w$ is the historical window. Regression task aims to learn a regression function $f_r(\cdot)$:

$$S_t = f_r(X). \tag{8}$$

---

[3]The detailed economic significance of $S_t$ is in Appendix A.2.

Table 1: Statistics of the stock investment risk benchmark.

| L | S | Train | Valid | Test |
|---|---|---|---|---|
| 1 | $[-2.00, -0.883]$ | 21,950 | 1,971 | 3,462 |
| 2 | $[-0.883, -0.407]$ | 101,209 | 12,792 | 22,895 |
| 3 | $[-0.407, 0.231]$ | 154,400 | 23,025 | 41,616 |
| 4 | $[0.231, 1.42]$ | 86,275 | 14,765 | 35,859 |
| 5 | $[1.42, 4.00]$ | 19,366 | 2,047 | 5,968 |
| total | - | 383,200 | 54,600 | 109,800 |

According to the distribution of $S_t$ in Figure 2, we derive the categorical risk level $L_t$:

$$L_t = \begin{cases} 1, & S_t \leq P_5 \\ 2, & P_5 < S_t \leq P_{30} \\ 3, & P_{30} < S_t \leq P_{70}, \\ 4, & P_{70} < S_t \leq P_{95} \\ 5, & P_{95} < S_t \end{cases} \tag{9}$$

where $L_1$, $L_2$, $L_3$, $L_4$ and $L_5$ are very low, low, medium, high, and very high risk levels, respectively.[4] Classification task aims to learn a classification function $f_c(\cdot)$:

$$L_t = f_c(X). \tag{10}$$

The two tasks are empirically hard due to implicit periodic information and correlation, where the regression is somehow difficult to be resolved compared to classification. Table 1 shows statistics of stock investment risk benchmark.[5] The dataset is split into training, validation, and test sets with a ratio of 7:1:2 in chronological order.

# 4 Models

In this section, we introduce our multi-view model architecture. As shown in Figure 3, the model architecture is composed of **LLM-based Time-Series Encoder (L-TSE)**, **Visual Time-Series Encoder (V-TSE)** and **Fusion Module**. L-TSE and V-TSE are used to extracts short-periodic and long-periodic information, respectively, which are fed to fusion model to obtain a comprehensive representation.

## 4.1 L-TSE

As shown in the bottom of Figure 3, the L-TSE branch consists of patch construction, patch reprogramming, and feature extraction. We adopt GPT2 model (Radford et al., 2019) as L-TSE backbone to extract short-periodic pattern from stock fundamental data.

**Patch Construction** We segment the input sequence of stock features $X \in \mathbb{R}^{w \times 1}$ into $X_p \in \mathbb{R}^{P \times l_p}$ overlapped patches (Wu et al., 2022). The total number of patches is calculated as:

$$P = \left\lfloor \frac{w - l_p}{s} \right\rfloor + 2 \tag{11}$$

where $l_p$, $s$ and $w$ denote the patch length, stride length and the historical window, respectively. Each patch is then embedded into a $d_m$ dimensional vector $X_d \in \mathbb{R}^{P \times d_m}$ using a linear projection layer.

---

[4]The categorization refers to Dowd & Blake (2006).

[5]In this work, we focus on single-channel time-series problem.

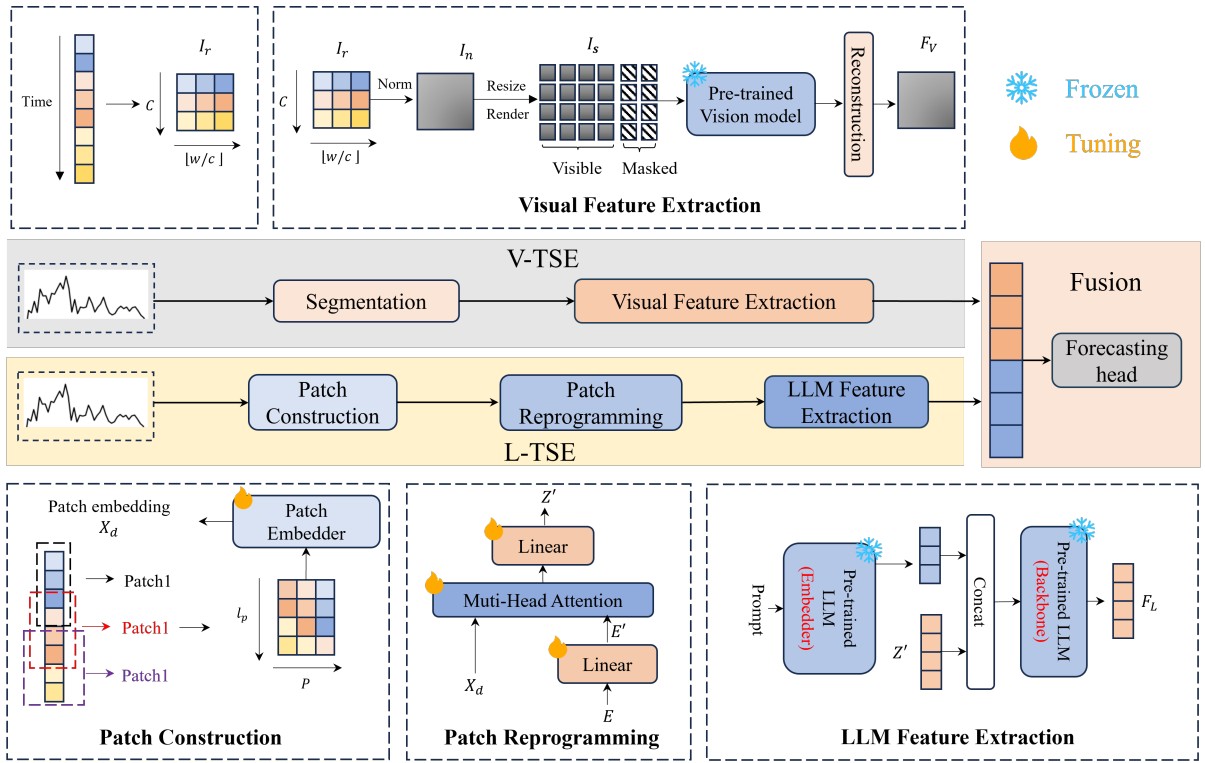

Figure 3: The multi-view modeling.

**Patch Reprogramming** To align patch embeddings with the representation space of the LLM, we re-program patch embeddings through a prototype-based multi-head cross attention mechanism and project to align the LLM hidden dimensions. Directly interacting patch embeddings with the full LLM vocabulary embedding $E \in \mathbb{R}^{V \times D_l}$ computationally inefficient due to the large vocabulary size $V$. To alleviate this issue, we project $E$ into a compact set of text prototypes $E' \in \mathbb{R}^{V' \times D_l}$, where $V' \ll V$ (Jin et al., 2023). The patch reprogramming is define as:

$$Z = \text{MHA}(Q, K, V), \tag{12}$$

$$Q = X_d W^Q, K = E'W^K, V = E'W^V, \tag{13}$$

$$Z' = \text{Linear}(Z), \tag{14}$$

where $W^Q \in R^{d_m \times d_m}$ is the linear projection for the queries, and $W^K, W^V \in R^{D_l \times d_m}$ are the projections for the keys and values. $Z' \in R^{P \times D_l}$ is the aligned patch embedding. MHA($\cdot$) is the multi-head cross attention operator and Linear($\cdot$) is a linear projection layer.

**LLM Feature Extraction** We design prompts based on the meta data of task description,[6] which is concatenated with patch reprogramming embedding $Z'$ as the input to the LLM to obtain LLM feature $F_L$:

$$e = \text{Embedder}(o), \tag{15}$$

$$F_L = \text{Backbone}([e : Z'])[P_l :], \tag{16}$$

where $o$ is the prompt. $e \in \mathbb{R}^{P_l \times D_l}$ is the prompt embedding by Embedder($\cdot$). $F_L \in \mathbb{R}^{P \times D_l}$ is LLM feature by Backbone($\cdot$) for stock investment risk forecasting.

---

[6]The detailed prompt design is in Figure 6 (a).

### 4.2 V-TSE

As shown in the top of Figure 3, the V-TSE consists of segmentation and visual feature extraction. We employ a vision backbone initialized with a pre-trained MAE model (He et al., 2022), which enables structured representations from stock data.

**Segmentation** We propose to segment $X$ into $\lfloor w/c \rfloor$ subsequences of length $c$ (Wu et al., 2022). We segment $X$ into 2D matrix $I_r \in R^{c \times \lfloor w/c \rfloor}$, which enables the joint modeling of intra-period variations and inter-period dependencies across the same temporal phase.

**Visual Feature Extraction** To align the input of pre-trained MAE model, we normalize 2D $I_r$ to get $I_n$:

$$I_n = r \cdot \frac{I_r - \mu(I_r)}{\sigma(I_r)}, \tag{17}$$

where $r$ is a hyperparameter. $\mu(\cdot)$ is mean operator and $\sigma(\cdot)$ is standard deviation operator.

We resize $I_n$ to obtain visiable parts $I_v \in \mathbb{R}^{H \times W_v}$ to match the height and the number of visible patch columns, which is concatenated with a masked placeholder $I_m \in \mathbb{R}^{H \times W_m}$ as follows:

$$I_v = \text{RESIZE}(I_n), \tag{18}$$

$$I_f = [I_v; I_m], \tag{19}$$

where $I_f$ is rendered to be $I_s \in \mathbb{R}^{H \times W \times 3}$ that is fed to pre-trained MAE. Finally, we reconstruct the output of the MAE to obtain the visual feature $F_V \in \mathbb{R}^{H \times W \times 3}$ as follows:

$$F_V = \text{RECONSTRUCT}(\text{MAE}(I_s)), \tag{20}$$

where $\text{RESIZE}(\cdot)$ is an image rescaling operation that adjusts the spatial resolution of the input image and $\text{RECONSTRUCT}(\cdot)$ is an unpatchifying operation that maps the patch space back to the image space.

### 4.3 Fusion Module

Given the representations of $F_L$ from the L-TSE and $F_V$ from the V-TSE, we project them into time-series domain to obtain $F'_L \in \mathbb{R}^{\Delta d \times 1}$ and $F'_V \in \mathbb{R}^{\Delta d \times 1}$, which are concatenated to obtain unified representation $F$:

$$F = [F'_L; F'_V] \in \mathbb{R}^{2\Delta d \times 1}. \tag{21}$$

The fused feature $F$ is fed into a regression head and a classification head for forecasting stock value $\hat{S}_t$ and the risk level $\hat{L}_t$, respectively.

## 5 Experiments

We carry out the experiments on our proposed stock investment risk forecasting benchmark of classification task and regression task in single stock feature settings.

### 5.1 Settings

**Baselines** We compare our method with several representative time series models, including LLM-based methods TIME-LLM (Jin et al., 2023), S2IP-LLM (Pan et al., 2024), LLM-TPF (Pan et al., 2025) and the image-based methods VisionTS (Chen et al., 2024) and traditional method LSTM(Graves, 2012). LLM-based and image-based baseline models are trained on our dataset using their reported settings. LSTM model consists of 2 layers with a hidden size of 64, and is trained for 10 epochs with a learning rate of 1e-4.

**Data** Experiments are conducted on the investment risk forecasting dataset consisting of 200 stocks covering the period from 1/1/2013 to 31/12/2023 using 6 different stock features, including adjusted price, closing price, low price, open price, trading volume, and high price.

Table 2: Results across our models and various baselines on regression task and classification task. Aclose, Close, Low, Open, Vol and High are adjusted close price, close price, low price, open price, trade volume and high price as stock features, respectively. The best scores are **bold** and the second best are underline.

| | | Regression Task | | | | | | | | | | | | Classification Task | | | | | | | | | | | |
|---|---|---|---|---|---|---|---|---|---|---|---|---|---|---|---|---|---|---|---|---|---|---|---|---|---|
| | | Ours | | Time-LLM | | S2IP-LLM | | LLM-TPF | | VisionTS | | LSTM | | Ours | | Time-LLM | | S2IP-LLM | | LLM-TPF | | VisionTS | | LSTM | |
| Metric | | MSE | MAE | MSE | MAE | MSE | MAE | MSE | MAE | MSE | MAE | MSE | MAE | Acc | F1 | Acc | F1 | Acc | F1 | Acc | F1 | Acc | F1 | Acc | F1 |
| Aclose | 96 | **0.390** | **0.466** | 0.527 | 0.545 | 0.532 | 0.545 | 0.548 | 0.548 | 0.524 | 0.548 | 0.528 | 0.544 | **0.453** | **0.411** | 0.382 | 0.243 | 0.436 | 0.377 | 0.380 | 0.215 | 0.381 | 0.259 | 0.382 | 0.226 |
| | 192 | **0.383** | **0.471** | 0.442 | 0.490 | 0.528 | 0.545 | 0.563 | 0.554 | 0.517 | 0.538 | 0.521 | 0.543 | **0.455** | **0.411** | 0.413 | 0.371 | 0.436 | 0.377 | 0.377 | 0.210 | 0.401 | 0.308 | 0.381 | 0.226 |
| | 336 | 0.479 | **0.519** | 0.472 | 0.526 | 0.530 | 0.545 | 0.550 | 0.549 | 0.539 | 0.559 | 0.517 | 0.541 | **0.429** | 0.378 | 0.408 | **0.385** | 0.398 | 0.273 | 0.379 | 0.217 | 0.390 | 0.311 | 0.382 | 0.235 |
| Close | 96 | **0.390** | **0.466** | 0.527 | 0.545 | 0.537 | 0.545 | 0.541 | 0.547 | 0.531 | 0.549 | 0.520 | 0.541 | **0.444** | **0.396** | 0.392 | 0.269 | 0.437 | 0.375 | 0.366 | 0.244 | 0.378 | 0.251 | 0.389 | 0.259 |
| | 192 | **0.388** | **0.470** | 0.461 | 0.517 | 0.528 | 0.545 | 0.553 | 0.550 | 0.529 | 0.542 | 0.521 | 0.543 | **0.456** | **0.414** | 0.397 | 0.314 | 0.447 | 0.402 | 0.380 | 0.215 | 0.396 | 0.310 | 0.381 | 0.226 |
| | 336 | **0.431** | **0.487** | 0.458 | 0.511 | 0.527 | 0.545 | 0.552 | 0.549 | 0.520 | 0.545 | 0.517 | 0.541 | **0.444** | **0.412** | 0.399 | 0.368 | 0.410 | 0.332 | 0.379 | 0.211 | 0.389 | 0.304 | 0.382 | 0.235 |
| Low | 96 | **0.393** | **0.466** | 0.524 | 0.547 | 0.530 | 0.545 | 0.558 | 0.551 | 0.532 | 0.551 | 0.520 | 0.541 | **0.457** | **0.415** | 0.400 | 0.298 | 0.432 | 0.364 | 0.380 | 0.211 | 0.377 | 0.262 | 0.384 | 0.247 |
| | 192 | **0.459** | 0.516 | 0.468 | **0.506** | 0.534 | 0.545 | 0.554 | 0.551 | 0.530 | 0.546 | 0.523 | 0.541 | **0.443** | **0.394** | 0.417 | 0.315 | 0.439 | 0.385 | 0.377 | 0.209 | 0.393 | 0.305 | 0.385 | 0.244 |
| | 336 | **0.478** | **0.520** | 0.528 | 0.536 | 0.528 | 0.545 | 0.559 | 0.552 | 0.526 | 0.548 | 0.523 | 0.541 | **0.477** | **0.422** | 0.352 | 0.291 | 0.383 | 0.252 | 0.375 | 0.214 | 0.388 | 0.306 | 0.379 | 0.212 |
| Open | 96 | **0.427** | **0.483** | 0.525 | 0.545 | 0.530 | 0.545 | 0.545 | 0.548 | 0.532 | 0.548 | 0.521 | 0.541 | **0.456** | **0.409** | 0.382 | 0.244 | 0.429 | 0.376 | 0.379 | 0.228 | 0.378 | 0.254 | 0.385 | 0.252 |
| | 192 | 0.499 | 0.538 | 0.449 | 0.499 | 0.530 | 0.545 | 0.548 | 0.548 | 0.527 | 0.553 | 0.523 | 0.541 | **0.462** | **0.423** | 0.427 | 0.340 | 0.429 | 0.376 | 0.379 | 0.212 | 0.393 | 0.310 | 0.384 | 0.237 |
| | 336 | 0.528 | **0.545** | 0.523 | 0.546 | 0.528 | 0.545 | 0.553 | 0.550 | 0.528 | 0.549 | 0.523 | 0.541 | **0.443** | **0.413** | 0.425 | 0.398 | | | 0.379 | 0.209 | 0.390 | 0.310 | 0.382 | 0.236 |
| Vol | 96 | 0.508 | 0.537 | **0.496** | 0.535 | 0.530 | 0.545 | 0.557 | 0.551 | 0.530 | 0.548 | 0.501 | 0.533 | 0.383 | 0.234 | **0.408** | **0.306** | 0.396 | 0.266 | 0.387 | 0.237 | 0.379 | 0.228 | 0.392 | 0.252 |
| | 192 | 0.520 | 0.539 | 0.492 | 0.535 | 0.526 | 0.545 | 0.550 | 0.550 | 0.542 | 0.550 | 0.495 | 0.532 | 0.391 | **0.281** | 0.380 | 0.223 | **0.400** | 0.277 | 0.385 | 0.226 | 0.378 | 0.256 | 0.394 | 0.256 |
| | 336 | 0.511 | 0.533 | 0.538 | 0.544 | 0.529 | 0.545 | 0.552 | 0.549 | 0.533 | 0.542 | **0.499** | 0.534 | 0.385 | 0.297 | 0.392 | 0.253 | 0.392 | 0.247 | 0.379 | 0.310 | 0.387 | 0.301 | **0.398** | 0.270 |
| High | 96 | **0.395** | **0.469** | 0.531 | 0.545 | 0.534 | 0.545 | 0.552 | 0.549 | 0.536 | 0.551 | 0.521 | 0.541 | **0.451** | **0.406** | 0.391 | 0.256 | 0.433 | 0.363 | 0.380 | 0.214 | 0.377 | 0.261 | 0.386 | 0.253 |
| | 192 | **0.397** | **0.476** | 0.521 | 0.553 | 0.527 | 0.545 | 0.556 | 0.553 | 0.532 | 0.544 | 0.523 | 0.541 | **0.450** | **0.408** | 0.431 | 0.385 | 0.441 | 0.392 | 0.380 | 0.215 | 0.392 | 0.300 | 0.385 | 0.244 |
| | 336 | 0.528 | 0.548 | **0.431** | **0.492** | 0.530 | 0.545 | 0.553 | 0.549 | 0.533 | 0.549 | 0.523 | 0.541 | 0.425 | 0.378 | **0.435** | **0.421** | 0.413 | 0.327 | 0.379 | 0.210 | 0.387 | 0.378 | 0.382 | 0.226 |

**Implementation Details**   All experiments are conducted on a single NVIDIA RTX 4090 GPU with 24GB of memory. The historical window is selected from $\{96, 192, 336\}$ and the forecasting horizon for stock investment risk is $\Delta d = 30$. In the patch construction, we set path length $l_p = 16$ and stride length $s = 8$ to construct patch and we set the hyperparameter $d_m = 512$. In the segmentation, we set subsequence length $c = 7$ inspiring by the inherent periodic patterns in stock market. In the visual feature extraction, the hyperparameter $r$ is fixed at 0.4. We use GPT-2-base as the backbone for the language model and MAE-base as the backbone for the vision model. The output dimensions of GPT-2 and MAE are denoted as $D_l = D_v = 768$.

**Metrics**   We evaluate regression performance using Mean Squared Error (MSE) and Mean Absolute Error (MAE), while classification performance is measured by accuracy and F1 score.

## 5.2   Results

As shown in Table 2, all results are obtained by averaging the evaluation metrics across all 200 stocks. Our model achieves superior performances on both regression task and classification task, demonstrating a clear overall advantage. Compared with LLM-based forecasting approaches and vision-based method, our method demonstrates clear performance gains, highlighting the limitation of relying single view on long-periodic patterns or short-periodic patterns for stock risk forecasting.

**Regression Task Results**   In the regression task, our model achieves the best performance on adjusted price, close price, low price, and high price across all historical windows. In particular, the best performance (MSE = 0.390, MAE = 0.466) is obtained when using the adjusted price as the stock feature with a historical window of 96. Compared with LLM-based, vision-based and traditional method baselines, including Time-LLM, S2IP-LLM, LLM-TPF, VisionTS and LSTM, our approach achieves significant improvements, outperforming their best results in all experiment settings by 11.8%, 27.9%, 24.6% and 21.8% based on MSE scores.

**Classification Task Results**   In the classification task, our model achieves the best performance. Our model with low price as stock features and historical window of 96 achieves the best performance of 0.477 accuracy and 0.422 F1. Compared other methods, our approach achieves significant improvements, outperforming their best results in all experiment settings by 0.48%, 25.5%, 36.5%, 11.9% and 56.7% based on F1 scores.

Table 3: Results of the ablation study on L-TSE and V-TSE. The best results are **bold**.

| Components | | Historical Windows | | | | | |
|:---:|:---:|:---:|:---:|:---:|:---:|:---:|:---:|
| L-TSE | V-TSE | 96 | | 192 | | 336 | |
| Regression Task | | | | | | | |
| | | MSE | MAE | MSE | MAE | MSE | MAE |
| ✓ | ✗ | 0.521 | 0.545 | 0.442 | 0.490 | **0.472** | 0.526 |
| ✗ | ✓ | 0.524 | 0.548 | 0.517 | 0.538 | 0.539 | 0.559 |
| ✓ | ✓ | **0.390** | **0.467** | **0.383** | **0.471** | 0.479 | **0.519** |
| Classification Task | | | | | | | |
| | | Acc | F1 | Acc | F1 | Acc | F1 |
| ✓ | ✗ | 0.382 | 0.243 | 0.413 | 0.371 | 0.408 | **0.385** |
| ✗ | ✓ | 0.380 | 0.215 | 0.401 | 0.308 | 0.390 | 0.311 |
| ✓ | ✓ | **0.454** | **0.411** | **0.455** | **0.411** | **0.429** | 0.378 |

Table 4: Results of multi-view modeling (adjusted close as stock features with historical window 96) with different feature fusion strategies. $F = [F_L; F_V]$ and $F = [F'_L; F'_V]$ are the concatenation fusion in the LLM space and the temporal space, respectively. $F = V \rightarrow L(F_L; F_V)$ and $F = L \rightarrow V(F_L; F_V)$ are the cross-attention fusion with $F_L$ and $F_V$ as the primary modality, respectively. The best results are **bold**.

| Feature fusion | Regression | | Classification | |
|:---|:---:|:---:|:---:|:---:|
| | MSE | MAE | Acc | F1 |
| $F = [F_L; F_V]$ | 0.370 | **0.454** | 0.449 | 0.402 |
| $F = V \rightarrow L(F_L; F_V)$ | **0.362** | 0.464 | 0.454 | 0.407 |
| $F = L \rightarrow V(F_L; F_V)$ | 0.390 | 0.465 | **0.465** | **0.430** |
| $F = [F'_L; F'_V]$ | 0.390 | 0.467 | 0.454 | 0.411 |

## 6 Analysis and Discussion

This section presents a comprehensive analysis of L-TSE, V-TSE, fusion methods, and the effects of stock features, historical window sizes, prompting strategies, and backbone models.

### 6.1 L-TSE and V-TSE

To validate the effectiveness of L-TSE and V-TSE module. We perform ablation study on the two modules. As shown in Table 3, under historical window of 96 and 192, using either module alone leads to a performance drop on both regression and classification tasks. However, under the historical window of 336, incorporating both modules does not yield further performance gains compared to using L-TSE alone. In both regression and classification task, L-TSE alone ahcieves higher performance than V-TSE alone.

These results provide insights into the roles of the two modules from multi-view perspective. Although both modules are derived from the same stock price sequence, L-TSE and V-TSE focus on different views of the data. L-TSE is used to capture the short-periodic pattern and V-TSE is used to capture the long-periodic pattern. L-TSE consistently outperforms V-TSE when used alone, indicating that short-periodic information play a more dominant role in stock risk forecasting. When the historical windows becomes long, both short-periodic and long-periodic patterns are affected by increased noise, leading to performance degradation.

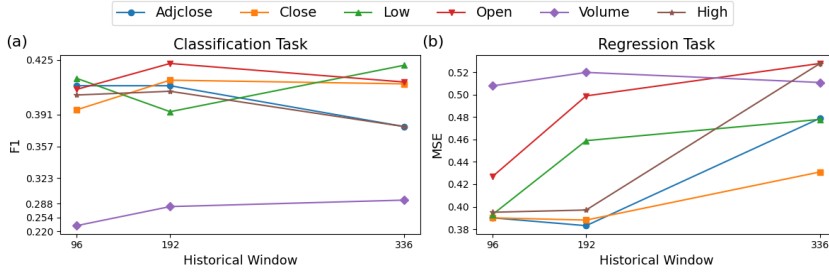
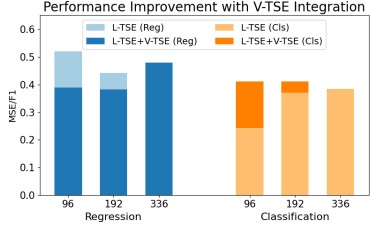

(a) Performance on regression and classification tasks.     (b) V-TSE Improvement.

Figure 4: (a) Performance comparison of regression and classification tasks across different historical windows and feature sets. (b) is the performance improvement of V-TSE with historical window 96 and adjusted close price as stock features. Blue bars denote regression results, while orange bars denote classification results. Light and dark colors indicate L-TSE and L-TSE+V-TSE, respectively.

## 6.2 Fusion Strategies

We conduct a systematic analysis of feature fusion (Sec. 4.3) by comparing to different fusion strategies, where we apply both cross attention and concatenation to implement the fusion module. As shown in Table 4, high-dimensional LLM feature space fusion outperforms the times-series space concatenation. Because high-dimensional LLM feature space fusion enables richer semantic interactions between modalities, allowing the model to capture more informative and complementary representations compared to simple concatenation in the time-series space.

## 6.3 Stock Features and Historical Windows

As shown in Figure 4a, model based on price-related features achieve better performance than trade volume based. Price-based feature provide direct information about stock return and risk-related price movements, which aligns the investment risk. In contrast, trading volume shows limited relevance to stock investment risk, since trading volume primarily reflect market attention and trading activity rather than the underlying investment risk of a stock. Trading volume contributes less discriminative information for investment risk forecasting compared to price-based features. This highlights that, for modeling investment risk, features directly related to price behavior are more reliable, while trading volume may serve better as a supplementary signal for stock investment risk forecasting.

Furthermore, different input features exhibit distinct temporal trends. In the classification task, close, open, and high follow a rising-then-declining pattern, while Low follows a declining pattern. In the regression task, adjclose and close follow a rising-then declining pattern, while low, open and high follow a declining pattern. It demonstrates that the optimal historical window length varies across different input features, indicating that the necessity of adaptive or feature-specific temporal modeling strategies.

As shown in Figure 4b, we observe that the contribution of V-TSE diminishes as the historical window size increases, and becomes negligible when the window size reaches 336. These results suggest that when the input sequence becomes longer, it may cause the input data to cover different market phases. This can lead to the presence of conflicting long-periodic patterns, which introduces noise into the model.

## 6.4 Prompt Strategies and Backbone Models

We investigate the effectiveness of different prompt strategies. As shown in Figure 6, to further investigate the impact of information granularity, we conducted experiments using three distinct prompt configurations: (a) statistical metrics (Ours), (b) financial metrics, and (c) hybrid combination of (a) and (b). The calculation of detailed financial metrics in Appendix A.3. As shown in Table 5, experimental results demonstrate that

Table 5: Performance comparison of different prompt strategies on regression and classification tasks. (a) statistical metrics (b) financial metrics (c) hybrid combination. The experimental results using the adjclose feature with a historical window of 96. The best results are **bold**.

| Prompt | Regression | | Classification | |
|---|---|---|---|---|
| | MSE | MAE | Acc | F1 |
| (a) | 0.402 | 0.469 | **0.454** | **0.413** |
| (b) | 0.390 | 0.467 | 0.454 | 0.411 |
| (c) | **0.376** | **0.464** | 0.448 | 0.406 |

Table 6: Performance comparison of different backbone models in L-TSE on regression and classification tasks. The experimental results using the adjclose feature with a historical window of 96. The best results are **bold**.

| Model | Regression | | Classification | |
|---|---|---|---|---|
| | MSE | MAE | Acc | F1 |
| Llama 7B | **0.371** | 0.467 | **0.462** | **0.433** |
| Bert-base | 0.390 | **0.463** | 0.438 | 0.421 |
| Ours (GPT-2) | 0.390 | 0.467 | 0.454 | 0.411 |

(c) enhances the precision of regression tasks, and (a) yields superior performance in classification tasks, reflecting the necessity of task-specific prompt strategies across different tasks.

We investigate the effectiveness of different backbone models in L-TSE. As shown in Table 6, the results show that upgrade the model backbone to Llama 7B leads to improvement in predictive accuracy.

## 7 Conclusion

In this paper, we construct a large-scale comprehensive stock investment risk dataset together with regression and classification tasks for benchmarking stock investment risk forecasting. To this end, we present a multi-view modeling for stock investment risk forecasting that integrates temporal information from stock prices with short-periodic feature extracted by LLM and long-periodic features from pre-trained vision models. Extensive experiments demonstrate that our method outperforms the competitive baselines, highlighting the benefit of multi-view modeling for the stock investment risk forecasting.

Despite its effectiveness, the current model still suffer from performance degradation in long financial sequences, as increasing historical windows introduce noisy. In future work, we plan to explore noise-filtering mechanisms to suppress less informative signals and improve representation quality.

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

# A Appendix

## A.1 The distribution of risk indicators

The distributions of volatility, maximum drawdown and beta are shown in Figure 5. Due to the heavy-tailed nature of financial risk indicators, extreme values are clipped using percentiles to preserve the overall distribution.

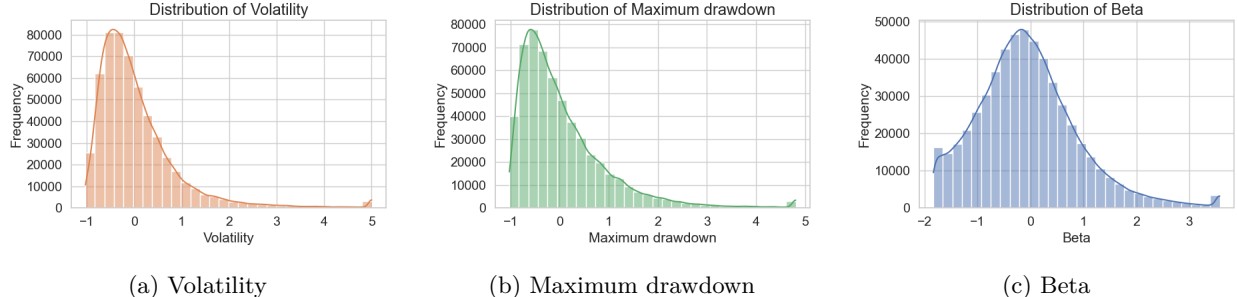

(a) Volatility          (b) Maximum drawdown          (c) Beta

Figure 5: Distributions of the three risk indicators after clipping the extreme 0.5% of values.

## A.2 Economic significance

The quantified investment risk $S_t$ integrates three dimensions of investment risk: return uncertainty, downside loss severity, and systematic market exposure. Specifically, volatility captures fluctuations in returns and reflects the uncertainty, maximum drawdown measures the magnitude of extreme losses and characterizes downside risk under adverse market conditions, and the beta coefficient quantifies a stock's sensitivity to overall market movements and represents systematic risk. $S_t$ offers a practical and actionable metric for risk monitoring, enabling the identification of potential market stress events and informing quantitative strategies.

## A.3 Prompt strategies

As shown in Figure 6, the prompt strategies include (a) statistical metrics, (b) Financial metrics and (c) hybrid combination of (a) and (b). Statistical metrics include maximum value, minimum value, median value and trend. Financial metrics include global volatility, mean growth intensity, distribution skew and signal persistence. The formulation of financial metrics as following:

- Global Volatility (CV) refer to the signals of non-stationarity. Its formulation as following:

$$CV = \frac{\sigma}{|\mu| + \epsilon} \tag{22}$$

  where $\mu$ is the mean value calculated from the input data. $\sigma$ is the standard deviation calculated from the input data. $\epsilon = 1e - 6$.

- Mean Growth Intensity captures the underlying momentum and trend consistency of asset price movements. Its formulation as following:

$$MGI = \frac{1}{n-1} \sum_{t=2}^{n} (x_t - x_{t-1}) \tag{23}$$

  where $x$ is the input data and $n$ is the number of input data.

- Distribution Skew used to identify extreme abnormal fluctuations. Its formulation as following:

$$DS = \frac{\mu - \tilde{x}}{\sigma + \epsilon} \tag{24}$$

- Signal Persistence quantifies the tendency of market trends to endure, reflecting the momentum or continuity of price dynamics. Its formulation as following:

$$SP = \frac{\sum_{t=1}^{n-1}(x_t - \mu)(x_{t+1} - \mu)}{\sum_{t=1}^{n}(x_t - \mu)^2 + \epsilon} \tag{25}$$

<|start_prompt|>
Dataset description: {description}

Task description: forecast the next {pred_len} steps given the previous {seq_len} steps information;

Input statistics:
min value:{min value},
max value:{max value},
median value: {median value},
the trend of input is {'upward' if trends[b] > 0 else 'downward'}
top 5 lags are : {lags_values_str}<|[object Object]|>

(a) Statistical metrics

<|start_prompt|>
Dataset description: {description}

Task description: forecast the next {pred_len} steps given the previous {seq_len} steps information;

Input statistics:
Global Volatility(CV):{cv_val},
Mean Growth Intensity:{mean_diff},
Distribution Skew: {skew_proxy},
Signal Persistence: {autocorr} <|[object Object]|>

(b) Financial metrics

<|start_prompt|>
Dataset description: {description}

Task description: forecast the next {pred_len} steps given the previous {seq_len} steps information;

Input statistics:
Global Volatility(CV):{cv_val},
Mean Growth Intensity:{mean_diff},
Distribution Skew: {skew_proxy},
Signal Persistence: {autocorr},
min value {min_values}, "
max value {max_values},
median value {median_values},
the trend of input is {'upward' if trends[b] > 0 else 'downward'} ,
top 5 lags are : {lags_values_str}<|[object Object]|>

(c) Hybrid combination

Figure 6: Prompt strategies include (a) statistical metrics, (b) financial metrics, (c) hybrid combination. The prompt includes dataset description, task description and input statistics. The highlighted parts in red are calculated from different input.

## A.4   Justification of Dataset Selection

In professional asset management, the SnP 500 serves as a standard investment universe where the primary objective is relative selection through risk-ranking. Our experiment design mimics the process of making decision in the real world. And the fluctuations of SnP 500 constituent stocks dominate the market and serve as the origin of risk transmission. Therefore, capturing the risk signals from these core equities is effectively equivalent to grasping the risk dynamics of the entire market.

## A.5   Normalization Process

To prevent data leakage, we normalized the features using only the mean and variance derived from the training data. The same parameters are subsequently applied to the validation and test sets. The normalization process as following:

$$\tilde{O}_{i,t} = \frac{O_{i,t} - \mu_O}{\sigma_O}, \tag{26}$$

where $O$ denotes the raw risk indicator. $\mu_O$ and $\sigma_O$ are computed from the training set.

