# OpenReview forum: "Multi-View Modeling for Stock Investment Risk Forecasting"
_TMLR — Rejected by TMLR_

### Review · Reviewer_MAEw · 2026-03-15

**Summary Of Contributions:**

The paper introduces a new stock dataset and a multi-view modelling framework for the stock price prediction problem, which is formulated as both a regression and a classification task. The model consists of an LLM component and a pre-trained vision model that are used to extract features from the original stock time-series.

Strengths
1. The paper presents a new dataset collected from the S&P 500, including 200 companies.
2.The experimental results, compared with four baselines, show that the proposed model outperforms the competing methods with respect to the MSE and MAE metrics.
3.	The paper presents several ablation studies on the effectiveness of the pre-trained models.

Weaknesses
1. The collected dataset is poorly presented. The paper does not describe how the 200 companies were selected from the S&P 500. In addition, the end date (31/12/2023) is outdated. Moreover, the paper does not conduct experiments on other datasets, such as FNSPID or QuanSIRA; therefore, the experimental results may be biased. The paper also does not explain how the data are split for the training and testing procedures. Furthermore, the dataset only contains stocks from the US market, which may limit the generalizability of the proposed framework.
2. Regarding the LLM used in the proposed framework, GPT-2 (2019), a relatively old pre-trained model, is used, which may limit the performance of the overall framework.
3. Regarding the performance measures, why are percentage-based error metrics such as MAPE or SMAPE not employed? In time-series prediction, especially stock price prediction, relative error metrics are very useful because the values of MAE and MSE can vary significantly depending on the currency of the stock price.

**Audience:**

Yes

**Audience Explanation:**

The use of pre-trained models in predict the stock price is interesting.

**Broader Impact Concerns:**

No significant ethical or broader impact concerns are identified.

**Claims And Evidence:**

No

**Claims Explanation:**

The dataset is unclear and the experimental results shoule be explained more clearly. For example, in Table 2, are the reported results averaged across the 200 stocks? How is this average computed, and is any weighting scheme used? In addition, the F1 scores are relative low, is it good enough for the practical application?

**Requested Changes:**

See above weaknesses.

---

> ### Author Response · Authors · 2026-04-04
> **About weakness**
>
> We sincerely thank the reviewers for their thoughtful feedback, which helps us improve the quality of our work.
>
> 1、Regarding limited datasets:
>
> We apologize for the lack of detail regarding our stock selection strategy in the original manuscript. In the revised version, we clarify that 200 companies are selected from the S&P 500 using a random sampling method. This approach is chosen to ensure a diverse and unbiased representation across different sectors and market capitalizations. To ensure full transparency, we will add a complete list of these 200 companies in the Appendix of the revised paper.
> We thank the reviewer for the suggestion to consider other financial datasets. Our risk quantification methodology is indeed inspired by the QuanSIRA dataset; however, we select a subset of risk metrics of QuanSIRA dataset to quantify stock investment risk. And FNSPID datasets is a dataset about stock price. Therefore, the QuanSIRA and FNSPID datasets are incompatible with our proposed model.
>
> 2、Regarding the use of an outdated LLM(GPT-2):
>
> We acknowledge that GPT-2 is a relatively older model compared to the latest LLMs. However, the selection of GPT-2 was a deliberate decision based on its computational efficiency. In real-world financial markets, inference latency and temporal sensitivity are as critical as predictive accuracy.
>
> 3、Regarding the choice of performance metrics:
>
> The target variable $S_t$ in our study is frequently distributed near or equal to zero.MAPE suffers from numerical explosion when the ground truth is near zero, while SMAPE becomes sensitive to errors.
>
> Response about:
>
> The dataset is unclear and the experimental results shoule be explained more clearly. For example, in Table 2, are the reported results averaged across the 200 stocks? How is this average computed, and is any weighting scheme used? In addition, the F1 scores are relative low, is it good enough for the practical application?
>
> We thank the reviewer for the valuable comments regarding the clarity of the dataset and experimental results.
>
> 1、The reported results in Table 2 are obtained by first computing predictions for each stock individually, and then averaging the evaluation metrics over all stocks. We will clarify this procedure in the revised manuscript.
>
> 2、Our task is a financial risk prediction problem, where the risk labels are constructed from multiple financial indicators, including volatility, maximum drawdown, and beta coefficient. This results in a fine-grained and continuous risk structure, making the classification task inherently challenging due to the noisy and overlapping nature of financial signals. And our method outperforms these baselines, demonstrating its effectiveness.

---

> ### Author Response · Authors · 2026-04-14
> **Regarding the use of an outdated LLM(GPT-2)**
>
> Thanks for review's suggestion. We run the experiment on different LLM. The results areas following: (feature=adj close, window size=96):
>
> The **Reg** represent the regression task and the **Cls** represent the classification task.
>
> | **Model**        | **Reg: MSE** | **Reg: MAE** | **Cls: Acc** | **Cls: F1** |
> | ---------------- | --------------------- | --------------------- | ------------------------------ | ------------------------ |
> | **Llama 7B**     | **0.3705**         | 0.4665             | **0.4621**                   | **0.4330**            |
> | **Bert-base**         | 0.3896              | **0.4629**         | 0.4382                      | 0.4211                  |
> | **Ours (GPT-2)** | 0.3896               | 0.4665               | 0.4535                        | 0.4112                   |
>
> The results show that upgrade the model backbone to Llama 7B leads to improvement in predictive accuracy. We will add the results to final revised version.

---

### Review · Reviewer_Yxk2 · 2026-03-15

**Summary Of Contributions:**

the paper introduce a multi-view framework for forecasting stock investment risk.
It provides two key contributions:

(1) a dataset for stock investment risk using data from 200 S&P 500 companies (2013–2023).

(2) a multi-view model with L-TSE for short term patterns and V-TSE for long term patterns

Strength:
- The authors target a specific problem in the finance domain. This experiment could have the practical application.

- The authors address a gap in the field by constructing a large-scale, standardized stock investment risk dataset. This includes quantifying risk using three distinct indicators: volatility, maximum drawdown, and the beta coefficient.

- The paper effectively argues that stock risk is composed of both short-periodic and long-periodic patterns. and idenitfy that fusing features in the temporal space is more effective than the high-dimensional LLM space.

Weakness
- While the combination is unique, the core components, GPT-2 and MAE, are existing foundations for a while. This might lead some to view the work as more of an application of existing tools than a fundamental algorithmic breakthrough.

- Using patching for time-series is a technique established in prior work like TimesNet , and reprogramming LLMs for non-text tasks has been explored in models like Time-LLM.

**Audience:**

Yes

**Audience Explanation:**

Individuals from financial domain will be specifically interested in this practice, especially experts in stock investment.

**Broader Impact Concerns:**

It doesn't have a broader impact statement. a thing could be added to clarify that the benchmark dataset is constructed from S&P 500 data from 2013 to 2023. This period is largely characterized by specific market regimes, There might be regime bias.

**Claims And Evidence:**

Yes

**Claims Explanation:**

The claims are mostly supported by experiment evidence.

- The paper proves the necessity of its dual-encoder architecture through comparative testing.

- Based on the most commonly used metrics, F1, MAE, MSE, the model achieves better accuracy in both regression and classification task.

- The clarity of the evidence is enhanced by a standardized, mathematical definition of the problem. It also provided standarded benchmark and clarified the architecture

But the long window noise problem and saturation point needs to be better addressed further.

**Requested Changes:**

- The authors note that performance significantly decreases as the historical window increases to 336 steps due to noise. A critical adjustment or analysis could be to implement, or propose a noise-filtering mechanism, or a gated fusion strategy to specifically handle the degradation observed in long-periodic patterns.

- The evidence shows a saturation point where adding the V-TSE module provides no benefit over L-TSE alone for a window of 336. The authors could provide a more rigorous explanation of why the visual long-periodic features fail at this specific threshold to ensure the multi-view claim is robust.

- The model’s success is heavily tied to price-related features, while trading volume shows "imited relevance. For a general recommendation, the authors could either justify why a stock risk model can largely ignore volume or show how the architecture could be adjusted to extract more discriminative signals from non-price data.

- The paper uses specific text prompts for the LLM. A brief sensitivity analysis on how different prompt formulations affect the patch reprogramming would demonstrate the stability of the L-TSE module.

---

> ### Author Response · Authors · 2026-04-04
> **About weakness**
>
> We sincerely thank the reviewers for their thoughtful feedback, which helps us improve the quality of our work.
>
> Weakness:
>
> 1、Regarding the limited novelty due to reliance on existing methods:
>
> We sincerely thank the reviewer for the insightful comments. We fully acknowledge that our work is built upon foundational techniques such as time-series patching (e.g., TimesNet) and LLM reprogramming (e.g., Time-LLM).  These works provided excellent starting points for our research, and we ensure that they are properly cited. However, we respectfully argue that our primary contribution is a fundamental architectural innovation that addresses the inherent limitations of current single-view models in investment risk forecasting.
>
> Requested changes:
>
> 1、Regarding the lack of mechanisms to address performance degradation in long sequences:
>
> We find these two perspectives (noise-filtering and gated fusion) enlightening. We fully agree that such mechanisms are critical for long financial sequences. Consequently, we will incorporate a detailed discussion of these strategies into the Future Work section of our revised manuscript.
>
> 2、Regarding the lack of explanation for the ineffectiveness of V-TSE at long windows:
>
> We sincerely thank the reviewer for this sharp and constructive observation. We agree that providing a more rigorous explanation for the saturation of the V-TSE module at the 336-step threshold is essential for the robustness of our multi-view claim.
> A clearer and more detailed analysis：At a length of 336, the 48 constituent periodic sequences (T=7) exhibit conflicting patterns due to market regime shifts. This introduces substantial noise into the V-TSE module, as it struggles to extract long-term information. This pattern interference causes the V-TSE to degrade.
>
> 3、Regarding the over-reliance on price features and limited use of volume information:
>
> We sincerely thank the reviewer for this insightful observation. The limited relevance of trading volume compared to price features is a key finding in our experiments, which can be justified by the mathematical nature of our target variables: The investment risk indicators are primarily constructed as functions of price sequences. This inherent numerical alignment ensures that price-based features achieve superior performance compared to based on trading volume.
>
> 4、Regarding the sensitivity of L-TSE to prompt formulation:
>
> We sincerely thank the reviewer for this constructive suggestion regarding the impact of different prompts. To ensure the overall scientific rigor of the manuscript, we have dedicated significant efforts during this rebuttal period to addressing the fundamental question raised by Reviewer 8GYt. While these experiments are currently underway, we require additional time to compile and verify the final results. We will incorporate the comprehensive findings into the final revised version.

---

> ### Author Response · Authors · 2026-04-14
> **Regarding the sensitivity of L-TSE to prompt formulation**
>
> Thanks for review's insight suggestion.
>
> **Regarding the sensitivity of L-TSE to prompt formulation:**
>
> To further investigate the impact of information granularity, we conducted experiments using three distinct prompt configurations: (1) Statistical metrics, (2) Financial metrics, and (3) a Hybrid combination of (1) and (2). The prompt structure as following:
>
> **(1)Statistical metrics(Identical to the format used in our manuscript):**
>
> <|start_prompt|>Dataset description: {description}
>
> Task description: forecast the next {pred_len} steps given the previous {seq_len} steps information;
> Input statistics:
>
> min value {min_values},
>
> max value {max_values},
>
> median value {median_values},
>
> the trend of input is {'upward' if trends[b] > 0 else 'downward'},
>
> top 5 lags are : {lags_values_str}<|<end_prompt>|>
>
> **(2)Financial metrics:**
>
> <|start_prompt|>Dataset description: {description}
>
> Task description: forecast the next {pred_len} steps given the previous {seq_len} steps information;
> Input statistics:
>
> Input statistics:
>
> Global Volatility(CV):{cv_val},
>
> Mean Growth Intensity:{mean_diff},
>
> Distribution Skew: {skew_proxy},
>
> Signal Persistence: {autocorr},
>
> **(3)Hybrid combination of (1) and (2):**
>
> <|start_prompt|>Dataset description: {description}
>
> Task description: forecast the next {pred_len} steps given the previous {seq_len} steps information;
> Input statistics:
>
> Input statistics:
>
> Global Volatility(CV):{cv_val},
>
> Mean Growth Intensity:{mean_diff},
>
> Distribution Skew: {skew_proxy},
>
> Signal Persistence: {autocorr},
>
> min value {min_values}, "
>
> max value {max_values},
>
> median value {median_values},
>
> the trend of input is {'upward' if trends[b] > 0 else 'downward'}, "
>
> top 5 lags are : {lags_values}<|<end_prompt>|>")
>
> **Global Volatility（CV）refer to the signals of non-stationarity. Its formulation as following:**
>
> $$CV = \frac{\sigma}{|\mu| + \epsilon}$$
> where $\mu$ is the mean value calculated from the input data. $\sigma$ is the standard deviation calculated from the input data.  $\epsilon=1e-6$
>
> **Mean Growth Intensity captures the underlying momentum and trend consistency of asset price movements. Its formulation as following:**
>
> $$MGI = \frac{1}{n-1} \sum_{t=2}^{n} (x_t - x_{t-1})$$
>
> where $x$ is the input data and $n$ is the number of input data.
>
> **Distribution Skew used to identify extreme abnormal fluctuations. Its formulation as following:**
>
> $$DS = \frac{\mu - \tilde{x}}{\sigma + \epsilon}$$
>
> **Signal Persistence quantifies the tendency of market trends to endure, reflecting the momentum or continuity of price dynamics. Its formulation as following:**
>
> $$SP = \frac{\sum_{t=1}^{n-1} (x_t - \mu)(x_{t+1} - \mu)}{\sum_{t=1}^{n} (x_t - \mu)^2 + \epsilon}$$
>
> The experiment results as following(feature=adj close, window size=96):
>
> The **Reg** represent the regression task and the **Cls** represent the classification task.
>
> | **Prompt** | **Reg: MSE** | **Reg: MAE** | **Cls: Acc** | **Cls: F1** |
> | --------- | -------------- | -------------- | ------------- | ------------- |
> | (1)       | 0.4024   | 0.4690  | **0.4542** | **0.4134** |
> | (2)       | 0.3896        | 0.4665  | 0.4535       | 0.4112        |
> | (3)       | **0.3762**   | **0.4641**  | 0.4475    | 0.4062    |
>
> Experimental results demonstrate that incorporating  financial metrics (3) enhances the precision of regression tasks, and relying solely on statistical metrics (1) yields superior performance in classification tasks, reflecting a distinct task-specific adaptability across different feature dimensions.

---

### Review · Reviewer_8GYt · 2026-03-22

**Summary Of Contributions:**

The paper proposes stock risk forecasting benchmark and a multi-view framework combining LLM based Times Series Encoder (L-TSE) for short-periodic patterns and visual time series encoder (V-TSE) for long-periodic patterns. The problem is important and underexplored. However, there are major concerns in the paper that undermines its validity:
1) Drawdown formula doesn't incorporate temporal ordering for max and min, also the dataset shows the formula used was max/min -1 rather than what is stated in the paper.
2) Severe drift in risk levels between train and test splits, (based on table 1, for L = 1, train has 6.8% population and test has 0.3%).
3) Consistent performance regression as historical window increases (which contradicts the motivation for the V-TSE component).

**Audience:**

Yes

**Audience Explanation:**

Yes, a correctly constructed benchmark and multi-view framework for stock risk forecasting would be a genuinely valuable contribution to the TMLR community.

**Broader Impact Concerns:**

The model is trained exclusively on S&P 500 stocks (the largest, most liquid, relatively lower-risk companies in the market). Deploying this in real financial decision-making on a broader investment universe, where genuine high-risk stocks exist, could produce unreliable risk assessments. Additionally, S_t​ is a research proxy with no standard financial interpretation and should not be treated as a validated risk measure for real investment decisions.

**Claims And Evidence:**

No

**Claims Explanation:**

The central claim of a valid benchmark is not supported due to the following major concerns:

1. Drawdown Formula Is Wrong:
The paper states Equation (4) which ignores the temporal ordering of high and low. One major issue with ignoring temporal ordering is that a stock that goes from $50-$100 in 30 days is treated the same as $100-$50, which is wrong from a risk management perspective. Also looking at the dataset the implementation consistently uses drawdown = Max/Min -1 across every stock tested (AAPL, AMD, Adobe).
For AAPL on 1/2/13 (window 1/3/13 to 2/14/13): max = 16.4767 and min = 13.3698. The paper's formula gives 0.1886, the dataset reports 0.2324. Since MD feeds directly into S_t, every regression target and classification label in the dataset is corrupted.

2. Performance Degrades with Longer Windows
Figure 4 shows that MSE increases and F1 decreases consistently as historical window grows. This directly contradicts the core motivation for V-TSE, which exists specifically to capture long-periodic patterns from longer sequences.

3. Severe Class Distribution Drift
Table 1, show severe class drift between train and test datasets for example when L = 1, train has 6.8% population and test has 0.3%.

4. S&P 500 Stocks Are Inherently Low Risk
The dataset is restricted to S&P 500 stocks from 2013 to 2023, these are the largest, most liquid, lower-risk companies in the market. Since, the training data doesn't include actual high risk companies in the entire investment universe, the results are hard to generalize.

5. Traditional Baselines Are Missing
Comparison with simple baselines are not provided. Without them the practical value of the proposed approach cannot be assessed.

6. If Labels Are Created Using Percentiles, Why Is a Model Needed?
The classification labels L_t are derived from static percentile thresholds of S_t. The model is therefore learning to mimic a deterministic ranking function. If these percentile-based labels are considered reliable ground truth, a simple lookup of the same statistics would suffice (raising a fundamental question about what does this the deep learning framework adds beyond overfitting to this labelling logic).

**Requested Changes:**

The authors should address the following concerns:
1) Fix MD formula and recompute S_t
2) Acknowledge and address class distribution drift and performance degradation on larger windows.
3) Add details of normalization (it is missing in the paper)
4) Add performance comparison with simpler baselines.
5) Explain why training on SnP dataset can be used to generalize stock risk forecasting on broader investment universe.

---

> ### Author Response · Authors · 2026-04-04
> **About weakness**
>
> We sincerely thank the reviewers for their thoughtful feedback, which helps us improve the quality of our work.
>
> 1、Regarding drawdown formula problem:
>
> We sincerely thank the reviewer for carefully examining the drawdown formulation and identifying this issue. Following the reviewer’s suggestion, we are correcting the drawdown computation to regenerate the dataset and re-run all experiments. Due to the volume of experiments, it takes additional ten days to finish.
>
> 2、Regarding the contradiction of V-TSE in capturing long-periodic patterns:
>
> V-TSE is designed to capture the internal long-periodic dependencies of the input data, which is independent of the input length. The reason for the performance drop in Figure 4 as the input length increases is that the long-periodic patterns of stocks change over time. Therefore, when the input sequence becomes longer, it may cause the input data to cover different market phases. This can lead to the presence of conflicting long-periodic patterns, which introduces noise into the model.
>
>
> 3、Regarding the class distribution drift:
>
> We sincerely thank the reviewer for pointing out this critical detail. In the real-world stock market, data distribution shift is a highly challenging inherent characteristic, as market risk signals and underlying dynamics vary significantly over time. We must candidly state that explicitly resolving this market drift problem is not the primary focus of this paper. Our core contribution is constructing a multi-view modeling framework to capture long- term and short-term periodicities to achieve accurate stock investment risk forecasting. Despite the challenges posed by data shift, our experiments still demonstrate the effectiveness of our method. We deeply appreciate the reviewer's valuable feedback, and we will explicitly add a discussion regarding this issue in the limitations section of the revised manuscript.
>
>
> 4、Regarding the limitation of datasets:
>
> We appreciate the reviewer’s perspective that SnP 500 constituents are considered 'lower-risk' relative to the entire investable universe. However, the investment risk is a relatively. In professional asset management, the SnP 500 serves as a standard investment universe where the primary objective is relative selection through risk-ranking. Our experiment design mimics the process of making decision in the real world. And the fluctuations of SnP 500 constituent stocks dominate the market and serve as the origin of risk transmission. Therefore, capturing the risk signals from these core equities is effectively equivalent to grasping the risk dynamics of the entire market.
>
>
> 5、Regarding the lack of traditional baselines:
>
> We sincerely thank the reviewer for this constructive feedback. We agree that including traditional baselines is crucial for comprehensively assessing the practical value and relative advantages of our proposed approach. However, due to the scale of the dataset and the experimental design, the computation workload is intensive. These baseline experiments are currently actively running. We commit to updating the revised version with these traditional baselines results to validate of our work.
>
> 6、Regarding the necessity of the model under percentile-based labeling:
>
> We sincerely thank the reviewer for raising this fundamental question. We would like to clarify a crucial temporal distinction that makes such a lookup impossible in practice. $S_t$ and $L_t$ are constructed using future data (t+1 to t+W). During real-time forecasting at time t, this future risk $S_t$ and $L_t$ is unobservable. Therefore, it is impossible to predict $S_t$ and $L_t$ through look up of the same statistical data.
>
> Regression tasks suffer from inherent 'boundary vulnerability'. A minor numerical error of just 0.01, if it occurs at the critical threshold of percentile partitioning, will result in misclassification. Therefore, we introduce the classification task to optimize the decision boundaries, which provides the model with much stronger robustness near these critical points.

---

> ### Author Response · Authors · 2026-04-14
> **The results of re-running experiments.（Part 1)**
>
> We have corrected a calculation error regarding the Maximum Drawdown and re-running all related experiments presented in the paper. The updated results as following:
>
> The main experiment results: **classification task**.
>
> **Classification Task (window size = 96)**
> | Metric    |  TIME-LLM (Acc/F1) |  S2IP-LLM (Acc/F1) |  LLM-TPF (Acc/F1) |  VisionTS (Acc/F1) |  LSTM (Acc/F1) |  **Ours** (Acc/F1) |
> | :-------- | :---------------: | :---------------: | :--------------: | :---------------: | :-----------: | :------------------------: |
> | Aclose |   0.3822/0.2428   |   0.4360/0.3769   |  0.3803/0.2146   |   0.3809/0.2586   | 0.3818/0.2258 |     **0.4535/0.4112**      |
> | Close     |   0.3920/0.2686   |   0.4373/0.3753   |  0.3656/0.2437   |   0.3782/0.2511   | 0.3892/0.2588 |     **0.4441/0.3957**      |
> | Low       |   0.4002/0.2982   |   0.4316/0.3636   |  0.3795/0.2107   |   0.3770/0.2623   | 0.3837/0.2469 |     **0.4568/0.4151**      |
> | Open      |   0.3816/0.2438   |   0.4287/0.3762   |  0.3795/0.2278   |   0.3785/0.2540   | 0.3853/0.2515 |     **0.4562/0.4090**      |
> | Volume    | **0.4077/0.3062** |   0.3958/0.2664   |  0.3868/0.2369   |   0.3791/0.2285   | 0.3918/0.2525 |       0.3831/0.2340        |
> | High      |   0.3913/0.2563   |   0.4329/0.3632   |  0.3802/0.2144   |   0.3770/0.2615   | 0.3856/0.2526 |     **0.4511/0.4056**      |
>
> **Classification Task (window size = 192)**
>
> | Metric | TIME-LLM (Acc/F1) | S2IP-LLM (Acc/F1) | LLM-TPF (Acc/F1) | VisionTS (Acc/F1) | LSTM (Acc/F1) | **Ours** (Acc/F1) |
> | :--- | :---: | :---: | :---: | :---: | :---: | :---: |
> | Aclose | 0.4132/0.3715 | 0.4360/0.3769 | 0.3767/0.2092 | 0.4007/0.3077 | 0.3813/0.2259 | **0.4547/0.4112** |
> | Close | 0.3970/0.3138 | 0.4473/0.4016 | 0.3804/0.2149 | 0.3965/0.3099 | 0.3817/0.2270 | **0.4562/0.4145** |
> | Low | 0.4168/0.3153 | 0.4395/0.3847 | 0.3774/0.2090 | 0.3934/0.3054 | 0.3817/0.2265 | **0.4427/0.3943** |
> | Open | 0.4265/0.3400 | 0.4287/0.3762 | 0.3789/0.2122 | 0.3926/0.3099 | 0.3843/0.2372 | **0.4618/0.4233** |
> | Volume | 0.3799/0.2232 | **0.4004**/0.2767 | 0.3848/0.2262 | 0.3775/0.2560 | 0.3939/0.2562 | 0.3909/**0.2806** |
> | High| 0.4311/0.3853 | 0.4408/0.3920 | 0.3800/0.2151 | 0.3918/0.3002 | 0.3848/0.2435 | **0.4505/0.4078** |
>
> **Classification Task (window size = 336)**
>
> | Metric | TIME-LLM (Acc/F1) | S2IP-LLM (Acc/F1) | LLM-TPF (Acc/F1) | VisionTS (Acc/F1) | LSTM (Acc/F1) | **Ours** (Acc/F1) |
> | :--- | :---: | :---: | :---: | :---: | :---: | :---: |
> | Aclose | 0.4084/0.3848 | 0.3981/0.2733 | 0.3787/0.2168 | 0.3898/0.3111 | 0.3818/0.2347 | **0.4293/0.3776** |
> | Close | 0.3987/0.3677 | 0.4100/0.3318 | 0.3791/0.2102 | 0.3891/0.3038 | 0.3821/0.2289 | **0.4437/0.4117** |
> | Low | 0.3517/0.2911 | 0.3833/0.2519 | 0.3745/0.2139 | 0.3883/0.3061 | 0.3794/0.2118 | **0.4470/0.4224** |
> | Open | 0.4246/0.3979 | 0.3811/0.2559 | 0.3788/0.2085 | 0.3900/0.3091 | 0.3820/0.2360 | **0.4428/0.4128** |
> | Volume | 0.3925/0.2534 | 0.3913/0.2469 | 0.3792/0.2342 | 0.3853/0.2638 | **0.3977**/0.2695 | 0.3851/**0.2974** |
> | High | **0.4351/0.4209** | 0.4133/0.3266 | 0.3792/0.2095 | 0.3869/0.3012 | 0.3820/0.2257 | 0.4247/0.3780 |
>
>
> The main experiment results: **regression task**.
>
> **Regression Task(window size = 96)**
>
> | Metric | TIME-LLM (MSE/MAE) | S2IP-LLM (MSE/MAE) | LLM-TPF (MSE/MAE) | VisionTS (MSE/MAE) | LSTM (MSE/MAE) | **Ours** (MSE/MAE) |
> | :--- | :---: | :---: | :---: | :---: | :---: | :---: |
> | **Aclose** | 0.5271/0.5448 | 0.5316/0.5452 | 0.5480/0.5485 | 0.5243/0.5477 | 0.5276/0.5436 | **0.3896/0.4665** |
> | **Close** | 0.5335/0.5442 | 0.5311/0.5451 | 0.5407/0.5467 | 0.5312/0.5486 | 0.5205/0.5407 | **0.3891/0.4655** |
> | **Low** | 0.5236/0.5468 | 0.5305/0.5451 | 0.5578/0.5509 | 0.5322/0.5511 | 0.5201/0.5405 | **0.3938/0.4658** |
> | **Open** | 0.5251/0.5447 | 0.5301/0.5450 | 0.5448/0.5480 | 0.5319/0.5483 | 0.5205/0.5408 | **0.4273/0.4826** |
> | **Volume** | **0.4957**/0.5352 | 0.5297/0.5449 | 0.5569/0.5510 | 0.5303/0.5478 | 0.5013/**0.5332** | 0.5075/0.5366 |
> | **High** | 0.5310/0.5451 | 0.5343/0.5455 | 0.5522/0.5490 | 0.5363/0.5512 | 0.5207/0.5411 | **0.3947/0.4687** |
>
> Because the limitation of the comment, other results will present in the next comment.

---

> ### Author Response · Authors · 2026-04-14
> **The results of re-running experiments.（Part 2)**
>
> **Regression Task(window size = 192)**
> | Metric | TIME-LLM (MSE/MAE) | S2IP-LLM (MSE/MAE) | LLM-TPF (MSE/MAE) | VisionTS (MSE/MAE) | LSTM (MSE/MAE) | **Ours** (MSE/MAE) |
> | :--- | :---: | :---: | :---: | :---: | :---: | :---: |
> | **Aclose** | 0.4419/0.4903 | 0.5283/0.5451 | 0.5634/0.5538 | 0.5171/0.5384 | 0.5215/0.5430 | **0.3825/0.4708** |
> | **Close** | 0.4609/0.5171 | 0.5276/0.5449 | 0.5526/0.5495 | 0.5292/0.5421 | 0.5227/0.5411 | **0.3880/0.4703** |
> | **Low** | 0.4679/0.5064 | 0.5341/0.5455 | 0.5540/0.5515 | 0.5299/0.5456 | 0.5228/0.5410 | **0.4586/0.5160** |
> | **Open** | **0.4494/0.4985** | 0.5298/0.5451 | 0.5479/0.5483 | 0.5268/0.5526 | 0.5226/0.5410 | 0.4991/0.5383 |
> | **Volume** | 0.4902/0.5349 | 0.5258/0.5447 | 0.5503/0.5496 | 0.5419/0.5501 | **0.4946/0.5322** | 0.5202/0.5394 |
> | **High** | 0.5213/0.5533 | 0.5266/0.5449 | 0.5565/0.5528 | 0.5324/0.5444 | 0.5226/0.5411 | **0.3971/0.4758** |
>
> **Regression Task(window size = 336)**
>
> | Metric | TIME-LLM (MSE/MAE) | S2IP-LLM (MSE/MAE) | LLM-TPF (MSE/MAE) | VisionTS (MSE/MAE) | LSTM (MSE/MAE) | **Ours** (MSE/MAE) |
> | :--- | :---: | :---: | :---: | :---: | :---: | :---: |
> | **Aclose** | **0.4721**/0.5261 | 0.5295/0.5450 | 0.5497/0.5485 | 0.5394/0.5585 | 0.5174/0.5413 | 0.4787/**0.5191** |
> | **Close** | 0.4581/0.5113 | 0.5271/0.5450 | 0.5522/0.5492 | 0.5201/0.5449 | 0.5227/0.5409 | **0.4312/0.4867** |
> | **Low** | 0.5278/0.5364 | 0.5280/0.5450 | 0.5588/0.5517 | 0.5262/0.5485 | 0.5228/0.5409 | **0.4779/0.5196** |
> | **Open** | 0.5229/**0.5456** | 0.5278/0.5450 | 0.5531/0.5502 | 0.5276/0.5492 | **0.5226**/0.5409 | 0.5281/0.5454 |
> | **Volume** | 0.5376/0.5439 | 0.5289/0.5447 | 0.5518/0.5493 | 0.5328/0.5420 | **0.4987**/0.5338 | 0.5106/**0.5325** |
> | **High** | **0.4309/0.4922** | 0.5297/0.5451 | 0.5526/0.5493 | 0.5333/0.5490 | 0.5226/0.5410 | 0.5284/0.5483 |
>
> **The results of ablation study on L-TSE and V-TSE(feature = adj close):**
>
> The **Reg** represent the regression task and the **Cls** represent the classification task.
> |  L-TSE   | V-TSE |   Window size=96    |   Window size=192   |   Window size=336   |
> | :------: | :---: | :-----------------: | :-----------------: | :-----------------: |
> | **Reg.** |       |   **(MSE / MAE)**   |   **(MSE / MAE)**   |   **(MSE / MAE)**   |
> |    ✓     |   ×   |   0.5211 / 0.5448   |   0.4419 / 0.4903   | **0.4721** / 0.5261 |
> |    ×     |   ✓   |   0.5243 / 0.5477   |   0.5171 / 0.5384   |   0.5394 / 0.5585   |
> |    ✓     |   ✓   | **0.3896 / 0.4665** | **0.3825 / 0.4708** |   0.4787 /**0.5191**   |
> | **Cls.** |       |   **(Acc / F1)**    |   **(Acc / F1)**    |   **(Acc / F1)**    |
> |    ✓     |   ×   |   0.3822 / 0.2428   |   0.4132 / 0.3714   | 0.4084 / **0.3848** |
> |    ×     |   ✓   |   0.3803 / 0.2146   |   0.4007 / 0.3077   |   0.3898 / 0.3111   |
> |    ✓     |   ✓   | **0.4535 / 0.4112** | **0.4547 / 0.4112** | **0.4293** / 0.3776 |
>
> **The results of different fusion method(feature = adj close, window size = 96):**
>
> | Feature Fusion Method | MSE (Reg)  | MAE (Reg)  | Accuracy (Cls) |  F1 (Cls)  |
> | :-------------------- | :--------: | :--------: | :------------: | :--------: |
> | $F = L \rightarrow V$ |   0.3902   |   0.4646   |   **0.4652**   | **0.4298** |
> | $F = V \rightarrow L$ | **0.3621** |   0.4641   |     0.4540     |   0.4070   |
> | $F=[F_L;F_V]$         |   0.3697   | **0.4538** |     0.4491     |   0.4022   |
> | $F = [F_L'; F_V']$    |   0.3896   |   0.4665   |     0.4535     |   0.4112   |
>
> After rectifying the calculation error in Maximum Drawdown (MDD), we re-conducted the ablation study on different fusion strategies. The revised results reveal that the Cross-attention-based fusion method outperforms other alternatives. Given that the corrected results differ from those in the original version, we will comprehensively update the corresponding sections of the manuscript, including the experimental analysis, tables, and conclusion, to ensure the technical accuracy and integrity of our work.
>
> We have updated the relevant data in our code repository and provided additional information regarding the training logs to ensure the reproducibility of our work.

---

> > ### Author Response · Authors · 2026-04-14
> > **About requested change**
> >
> > **Regrading the requested change1:**
> > We revise the MDD  formulation as:
> > $$MDD(T) = \max_{\tau \in [0, T]} \left( \frac{\max_{t \in [0, \tau]} P(t) - P(\tau)}{\max_{t \in [0, \tau]} P(t)} \right)$$
> >
> > We have recomputed the $S_t$ and updated in our code repository.
> >
> > **Regrading the requested change2:**
> > We thank the reviewer for this constructive suggestion. We acknowledge the limitation of our current framework regarding data distribution shift. Effectively addressing distribution drift is a relatively independent research problem. This challenge is not the central focus of our study, but we recognize its critical importance and we will clarify this in the section of limitation and future work.
> >
> > **Regrading the requested change3:**
> > We thank the reviewer for this constructive suggestion. To prevent data leakage, we normalized the features using only the mean and variance derived from the training data. The same parameters were subsequently applied to the validation and test sets. We will add the detailed normalization process to revised paper.
> >
> > **Regrading the requested change4:**
> > We thank the reviewer for this insightful suggestion. We have included LSTM as a representative deep learning baseline to provide a more comprehensive performance comparison. The main experiment results show that.
> >
> > **Regrading the requested change5:**
> > In professional asset management, the SnP 500 serves as a standard investment universe where the primary objective is relative selection through risk-ranking. Our experiment design mimics the process of making decision in the real world. And the fluctuations of SnP 500 constituent stocks dominate the market and serve as the origin of risk transmission. Therefore, capturing the risk signals from these core equities is effectively equivalent to grasping the risk dynamics of the entire market.
> > We will add the explanation in the section of experiment setup to state the design and construction of our dataset.

---

> > > ### Comment · Reviewer_8GYt · 2026-04-14
> > > **Thanks for the updates**
> > >
> > > The additional analysis has sufficiently addressed all the issues and significantly enhanced the paper's quality.

---

### Decision · Action_Editor_EKZD · 2026-05-01

**Recommendation:** Reject

**Additional Comments:**

The initial reviews were mixed. While reviewers acknowledged the data collection effort, insightful ablation studies, and promising experimental results, they also raised substantial concerns regarding the presentation, evaluation methodology, and technical validity. During the rebuttal phase, the authors addressed some of these issues and introduced additional baseline comparisons (LSTM), alternative prompting strategies (Appendix A.3), and a full re-run of the experiments following the discovery of an error. These revisions have improved the paper’s overall quality, but the final recommendations remain divided.

Overall, the manuscript is still in an unpolished state, and the systemic issues outlined above are not easily resolved. Making these changes requires a level of revision that is not minor, and therefore necessitates a new review process. I encourage the authors to resubmit a revised version at a later date.

**Audience:**

No

**Audience Explanation:**

- While many works focus on predicting stock prices, forecasting investment risk is less common but valuable as part of a broader portfolio construction pipeline.
- The architecture combines two well-known LLM/vision models with a standard fusion approach; from a modeling perspective, this design is reasonable.
- The paper’s primary weakness lies in the formulation of the risk measure $S_t$, which averages three risk indicators with different units. As a result, $S_t$ aggregates variables on incompatible scales, lacks grounding in a pricing model, and has no clear semantic interpretation.
- The evaluation relies on a single, self-collected dataset with only six simple features and coarse temporal granularity. Moreover, the forecasts are based on just one of these features, making the setup far removed from common industry practice. The model captures neither feature nor stock dependencies, both of which are typically in the hundreds. The reported performance metrics are therefore of little practical relevance.
- A key missing component is an experiment demonstrating the advantage of predicting $S_t$ over traditional risk indicators, for example in terms of downstream portfolio performance.

The paper predicts a questionable target based on inadequate information, while the benefit of making these predictions remains unclear. In its current form the TMLR audience is unlikely to benefit from this analysis.

**Claims And Evidence:**

No

**Claims Explanation:**

- The paper provides insightful ablation studies analyzing the contributions of the two main modules (L-TSE and V-TSE), as well as the effects of window size, feature choices, prompts, and backbone architectures.
- The proposed method is evaluated against five baselines, including traditional approaches such as LSTM and more recent methods like Time-LLM. However, comparisons with stronger models (e.g., Moirai or Chronos) are missing.
- The risk indicators (beta, volatility, drawdown) initially contained multiple errors, necessitating a full re-run of experiments during the rebuttal phase. Although these issues have now been corrected, errors at such a fundamental level are concerning.
- The mathematical notation is inconsistent and several formulas (e.g., Eq. (6) and Eq. (11)) contain errors, including overloaded symbols, undefined variables, indexing mistakes, and dimension mismatches. Similar issues appear in Figure 3 (e.g., patch construction and V-TSE segmentation).
- The manuscript suffers from weak presentation, including grammatical issues (e.g., incomplete sentences), typographical errors (e.g., “ahcieves”), and incorrect use of financial terminology (e.g., “yield rate”).

Overall the paper is in an unpolished state and requires revisions in terms of clarity and precision.

**Resubmission Of Major Revision:**

The authors may consider submitting a major revision at a later time.